# The Role of Coupling Agents in the Mechanical and Thermal Properties of Polypropylene/Wood Flour Composites

Cecilia Zárate-Pérez [1] , Rodrigo Ramírez-Aguilar [1] , Edgar A. Franco-Urquiza [1,*] and Carlos Sánchez-Alvarado [2]

1    Advanced Manufacturing Department, Center for Engineering and Industrial Development (CIDESI), Carretera Estatal 200, km 23, Querétaro 76270, Mexico
2    Advanced Plastics Technologies (ADAPT), Avda, Industria Petroquímica #402 Lote 1, Manzana 12 Col. Parque Industrial, Querétaro 76220, Mexico
*    Correspondence: edgar.franco@cidesi.edu.mx

**Abstract:** This work is a collaborative effort between academia and industry to promote the development of new sustainable and profitable materials for manufacturing products. Incorporating wood flour particles (WF) in polypropylene (PP) grants environmental advantages in developing products that use renewable resources to manufacture PP/WF composites using the melt intercalation process. However, the interaction between a hydrophilic strengthening phase (wood flour) with a nonpolar polymer matrix (PP) is poor, resulting in deficient mechanical performance. This investigation details the use of graft and masterbatch coupling agents to evaluate their effects on mechanical parameters. The low compatibility between the constituents favors increasing the composites' thermal properties because the reinforcing phase acts as a nucleating agent. PP showed typical mechanical behavior, with a marked necking and a wide deformation capacity of approximately 180%. The mechanical behavior of the PP/WF composites revealed an elastic region followed by a termination after their yield point, shortening the stress–strain curves and reducing their ductility at strain values of approximately 2–4%. Graft coupling agents have better intermolecular performance with PP than masterbatch coupling agents. The modulus of elasticity of the composites increased to around 82% relative to PP. Processing methods influenced the thermal properties of the composites. The melt-blending process promoted molecular orientation, while injection molding erased the thermomechanical history of the extruded pellets. The melting temperature was similar in the composites, so there was no evidence of thermal degradation. The results showed that the coupling agents favor the crystallinity of the PP over tensile strength. SEM observations showed insufficient adhesion between the WF and PP, which promotes a reduction in stress transfer during tensile testing. The WF particles act as fillers that increase the stiffness and reduce the ductility of composites.

**Keywords:** graft coupling agents; WF particles; WF/PP composites; mechanical performance; thermal properties

## 1. Introduction

Over the last few years, the campaign for the care and protection of the environment has intensified through the sustainable development objectives set forth by the United Nations (UN). Part of these objectives focus on nations' ecological and economic ability to strengthen the development of composite materials using natural fillers [1–10]. Although using natural fillers is not as popular as using mineral or inorganic fillers [1,11–15], reusing forest or agricultural by-products has several advantages: low density, processing flexibility, acceptable specific mechanical properties, availability, and low cost.

Wood–plastic composites (WPC) reuse cellulose residues (wood flour, wood fibers, and cellulose fibers) to increase the specific mechanical properties of thermoplastics and reduce costs. WPCs are lightweight, recyclable, resistant to corrosion, have excellent

mechanical and impact properties, and do not harm the environment [16–20]. WPCs can now solve plastic and wood waste disposal problems, which implies using neat or recycled polymers. Commonly used thermoplastic polymers for WPCs are polyolefins, polyamides, or polystyrenes [18,21–23]. The polymer most widely used by industry and academia worldwide is polypropylene (PP) due to its easy processing, low cost, and suitable properties for various applications including automotive, commodities, electronics, and medical instruments. However, due to its high daily consumption and rapid disposal, large amounts of waste PP have accumulated. Therefore, compounding appears to be the more suitable technology for developing WPC.

Combining PP and wood flour in different filler concentrations can produce composite materials with improved mechanical properties. However, the main problem related to the use of wood flour is its low compatibility with most thermoplastics. Wood flour is a lignocellulosic-based material and consequently highly hydrophilic, while PP is inherently hydrophobic. This fact reduces PP–wood flour interactions, which ultimately results in poor mechanical performance and reduced thermal properties since the properties of WPC are highly dependent on compatibility and interfacial adhesion between wood and the surface polymer matrix [24–27].

Coupling agents allow for the chemical bonding of intrinsically different materials. They are essentially bifunctional groups that contain different types of molecules to increase the molecular interaction of the interface between polymer chains and fillers [20,28–31]. In this way, one group can have a chemical reaction or good compatibility with the polymeric matrix, while the other group can form chemical bonds with the reinforcement, thus coupling the two constituents. In this manner, the coupling agents improve interfacial adhesion and significantly increase the performance of the polymeric composite.

Diverse coupling agents are based on silanes, titanates, or zirconium aluminate. Additionally, graft copolymers are used as compatibilizers. Graft copolymers are macromolecules of two or more different chemical chains in which one chain (called a backbone) has multiple branches formed from macromolecular chains with a different chemical composition than that of the backbone [32–35].

The effectiveness of graft copolymers as coupling agents works by chemically improving the interfacial interactions between the polymer matrix and the lignocellulosic fibers. Matheus Poletto [36] prepared recycled PP composites reinforced with treated and untreated wood flour using different natural oils as additives. Poletto found that using natural oils as coupling agents improved interfacial adhesion between the wood flour and the polypropylene matrix. One of the composites presented mechanical and thermal properties similar to those promoted by polypropylene grafted with maleic anhydride. Hong et al. [36] used multi-monomer graft copolymers of polyethylene and polyethylene wax to compatibilize recycled polyethylene/wood flour composites. The authors detailed that the synergistic compatibilization between polyethylene and wood flour offers a robust interfacial interaction, while polyethylene wax complements the interfacial interaction by penetrating the cavities and capillaries of the wood particles. Du et al. [36] evaluated the effect of styrene-assisted maleic anhydride-grafted poly(lactic acid) (PLA-g-St/MAH) on the interfacial properties of wood flour/PLA biocomposites. PLA-g-St/MAH was synthesized by free radical melt grafting using styrene as comonomer and dicumyl peroxide as an initiator. The results indicated that the storage modulus, complex viscosity, equilibrium torque, and shear heat increased significantly. The mechanical properties of the composites also increased substantially after the addition of PLA-g-St/MAH. The compatibilization between polyethylene and wood flour was also achieved by other authors, who used maleic anhydride (MAH) and comonomer styrene (St) during the reactive extrusion process [36]. Harper et al. [36] also studied the PP/PP-g-MAH mixture and the corresponding pure PP. Their work on the blend suggested that the grafting of MAH led to a lower melting point of the PP component, probably due to the co-crystallization of PP and PP-g-MAH. Compared with the pure PP homopolymer, the dynamic storage modulus of the blend

increased slightly with the addition of a small amount of PP-g-MAH, while the mechanical damping decreased with the addition of PP-g-MAH.

This work aims to evaluate the effect of the addition of coupling agents on the mechanical and thermal properties of PP/WF composites by using grafted copolymers based on Poly(styrene–glycidyl methacrylate)-GMA- or Poly(styrene-co-methyl methacrylate-co-glycidyl methacrylate)-MMA-, which are synthesized by the nitroxide-mediated radical polymerization of styrene. The PP/WF composites were prepared through a melt-blending process.

## 2. Materials and Methods

### 2.1. Materials

Axlene® 35 PP homopolymer from INDELPRO (Tamaulipas, Mexico) with a density of 0.905 cm$^{-3}$ and melt flow rate of 35 g/10 min (190 °C, 2.16 kg according to ASTM D1238) was used as the polymer matrix. Arbocel® wood flour Grade C350 SR from JRS (Rosenberg, Germany) with an average particle size between 70–150 μm was used as the natural filler. Aerosil® 200 hydrophilic fumed silica from Evonik IM—Glenn Corp (Essen, Germany) with a specific surface area of 200 m$^2$/g was used as an additive to hinder moisture integration into the PP/wood flour composites. Grafted copolymers 5901 and terpolymers 5951 from AddiCo (Toluca, Mexico) with poly(styrene–glycidyl methacrylate) and poly(styrene-co-methyl methacrylate-co-glycidyl methacrylate), respectively, were used as coupling agents to evaluate the molecular compatibility between PP and wood flour. Another kind of modifier, coupling masterbatch CR-0218 from Ameriplas Inc. (Conroe, TX, USA), is a multi-faceted concentrate used for coupling wood to PE/PP and post-consumer PE/PP resins while providing compatibilization with polar polymers, coupling wood, minerals, or glass, or used in the dispersion of colors. The materials were kindly provided by ADAPT (Queretaro, Mexico).

### 2.2. Melt-Blending Procedure and Injection Molding

The melt-blending approach is a thermomechanical process that allows the polymer to be melted and mixed with additives and reinforcements through temperature, residence time, and shear stress. The PP/WF composites, containing a nominal 30 wt. % of WF and 3 wt. % of coupling agents, were prepared in a ZSK-18 MI COPERION (Stuttgart, Germany) twin-screw extruder (L/D = 48, D = 18 mm). The extruder contains nine temperature-controlled sections and a side feeder for dosing critical additives and reinforcements. Following the processing recommendations indicated in the datasheet, the suitable temperature profile was set from 140 °C in the feed section to 190 °C at the extrusion die. The twin-screw rotation speed was set at 80 rpm. PP and modifiers were fed in Section 1, while WF was added in Section 3 using the side feeder at 60 rpm. The residence time was measured as ~126 s for all composites. After extrusion, the PP/WF composites were cooled in a water bath and pelletized.

Dumbbell-like specimens (ASTM D6380 type V) of non-extruded PP and composites were injection-molded in a BabyPlast 6/12 (Molteno, Italy) microinjection machine. The temperature profiles used in the three zones during the injection process for all samples were 188 °C, 204 °C, and 214 °C, with a closed-mold pressure of 135 MPa.

It is essential to highlight that raw materials and PP/WF composites were dried at 90 °C for 5 h in an air oven before the melt-extrusion process and injection molding, as the presence of water may lead to significant modifications in the composite structure.

### 2.3. Experimental Methods

From PP and PP/WF composite pellets, microtome cut samples of approximately 80 μm in thickness were taken and analyzed using a Fourier transform infrared (FTIR) spectroscope from Perkin Elmer Frontier (Waltham, MA, USA). FTIR spectra were recorded by scanning the samples in frequencies 4000–500 cm$^{-1}$. The specimens were scanned 16 times in the transmittance mode at a resolution of 4 cm$^{-1}$.

Thermal behavior was analyzed using a TA Instruments Discovery DSC 250 (New Castle, DE, USA) differential scanning calorimeter. Pellets of non-extruded PP and PP/WF composites approximately 10 mg in weight were put in aluminum pans and thermally evaluated through a first heating scan from 30 to 220 °C with an isothermal step of 1 min, followed by a cooling scan with an isothermal step of 1 min and a second heating scan. All runs were conducted at a heating/cooling rate of 10 °C/min under a nitrogen atmosphere.

The crystallinity of the neat PP was calculated by the following equation [37]:

$$X_{DSC} = \frac{\Delta H_m}{\Delta H_{PP}} x * 100 \qquad (1)$$

while the percentage of crystallinity of the composites was determined by the following equation [38]:

$$X_{DSC} = \frac{\Delta H_m}{\Delta H_{PP}(1 - \varphi)} x * 100 \qquad (2)$$

where $\Delta H_m$ is the measured melting enthalpy, $\Delta H_{PP}$ is the theoretical melting enthalpy of a 100% crystalline PP [39], and $\varphi$ is the weight percentage of WF filler and coupling agent. The enthalpy of a completely crystalline PP was assumed to be 207 J/g.

The mechanical characterization of PP and PP/WF composites was performed using an Instron 647 (Norwood, MA, USA) universal testing machine, equipped with a load cell of 10 kN. Eight dumbbell-like specimens were tested in uniaxial tensile tests according to ISO-527. The type V specimens were tested at a 10 mm/min crosshead rate at room temperature (23 ± 2 °C). Mechanical parameters such as Young's modulus (E), yield strength ($\sigma_y$), and tensile strength at break ($\sigma b$) were calculated from the engineering stress–strain curves, and the nominal strain ($\varepsilon_t$) was determined using method A.

The failure surface of post-mortem tensile specimens was observed with a JEOL JSM 400 scanning electron microscope (SEM).

## 3. Results

The FTIR spectra of the WF and PP/WF composites are shown in Figure 1.

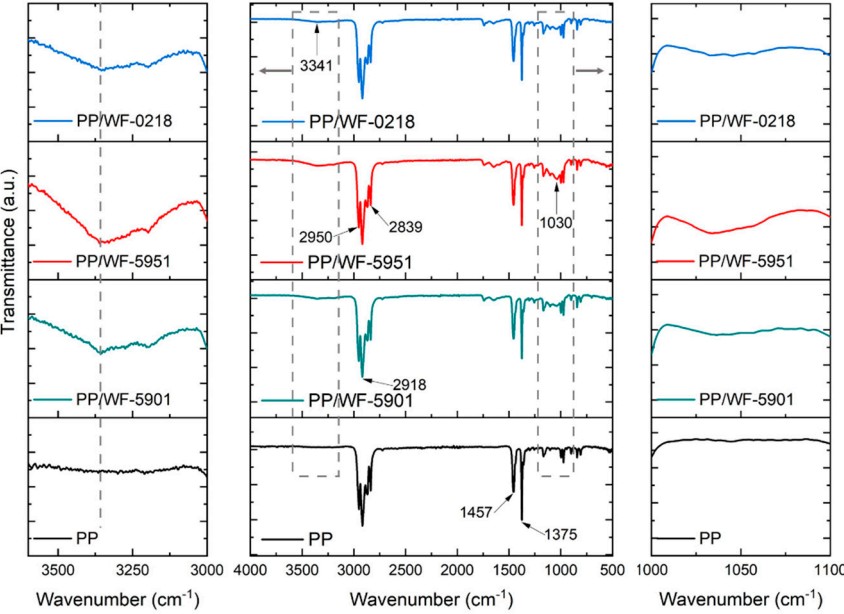

**Figure 1.** FTIR spectra of PP and PP/WF composites. Enlarged regions are to the left and right of the complete spectra.

The spectra of the PP/WF composites were very similar, with each displaying the characteristic band compositions of PP [40], which suggests the coupling agents were

composed of grafted PP-based copolymers. The FTIR spectrum of PP/WF composites showed the characteristic peaks of PP. The bands at 2950 cm$^{-1}$ and 2839 cm$^{-1}$ are related to CH$_3$ asymmetrical stretching and the band at 2918 cm$^{-1}$ is related to the CH$_2$ stretching peak. The methyl group umbrella mode (symmetric bending vibration mode of –CH$_3$ group) is present in the band at 1375 cm$^{-1}$ [41] and the peak located at 1457 cm$^{-1}$ is attributed to CH$_3$ asymmetric deformation [42–45].

Fingerprint absorption in the region from 600–1600 cm$^{-1}$ is typically assigned to major cell-wall components such as cellulose, hemicelluloses, and lignin [46,47], which is the range of the typical PP FITR spectrum. The region of 1060–1030 cm$^{-1}$ is more specific for the biomass and the bands around 1373 cm$^{-1}$ are mainly from PP [48].

It is possible to appreciate the most significant changes in the ranges 3341 cm$^{-1}$ and 1030 cm$^{-1}$ (the first inset dashed grey lines in Figure 1). The broad peak observed at 3341 cm$^{-1}$ is related to -OH stretching vibration on hydroxyl groups, which originates mainly from the cellulose of wood flour [49–53]. The zoom inset in Figure 1 shows that the intensity of this broad peak varies conspicuously, being slightly more intense in composites PP/WF-5901 and PP/WF-5951. The variation in the intensity of this PP/WF composite signal indicates the hydroxyl groups consumed in the modification process, as stated by other authors [54].

The signal in the PP/WF composites at around 1030 cm$^{-1}$ (second inset right in Figure 1) is assigned to C-O stretching in the primary alcohols of lignin in wood flour. Similar to the hydroxyl group changes, less-intense vibration characteristics are observed for the composites' spectra, indicating a reduction in the number of primary alcohols on the surface of the wood flour [54]. The shoulder reduction is conspicuous in the PP/WF-5901 and PP/WF-0218 composites. The observed changes are characteristic of a decrease in polarity on the surface of the wood flour, which benefits adequate mixing with polymers.

Figure 2 shows the stress vs. nominal strain engineering curves for neat PP and PP/WF composites.

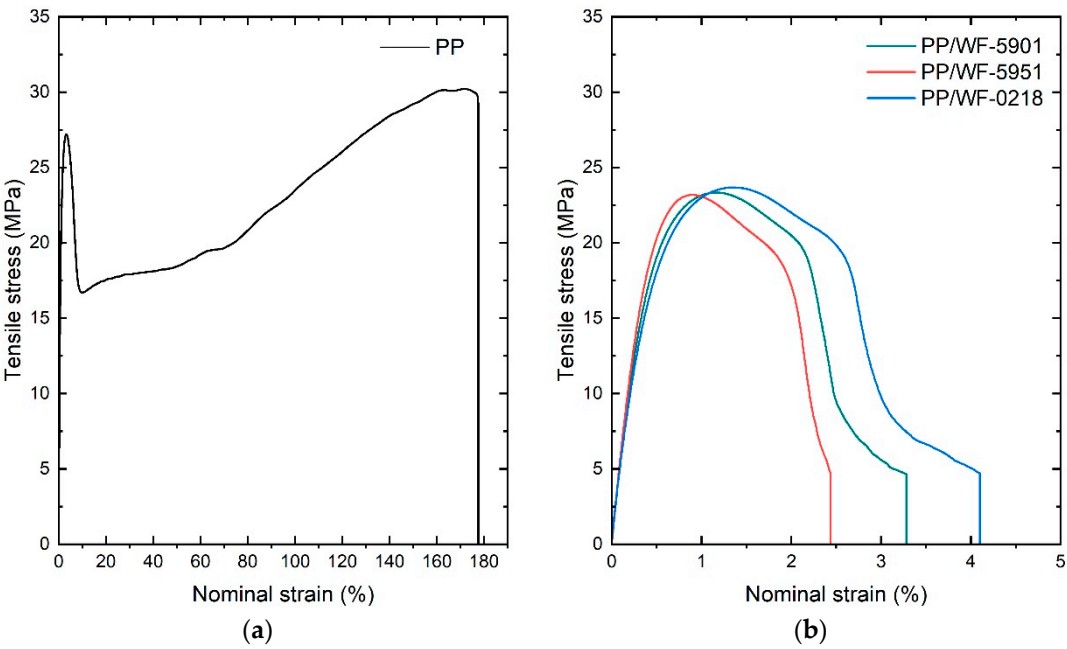

**Figure 2.** Stress–strain curves corresponding to (**a**) neat PP and (**b**) PP/WF composites.

It is possible to appreciate that the neat PP curve (Figure 2a) presents a typical behavior, with a very marked localized necking followed by a homogeneous deformation and strain hardening before its failure. PP showed a very high deformation of approximately 180%.

In the case of the PP/WF composites, the mechanical behavior turned out to be very different from that of neat PP, as observed in Figure 2b. The composites showed a linear

behavior, a plastic region without evident necking, and termination after their yield point, shortening the stress–strain curves, reducing their ductility, and reaching 2–4% strain values. Table 1 presents the values of the mechanical parameters obtained from the stress–strain curves of the analyzed materials.

**Table 1.** Mechanical parameters of neat PP and PP/WF composites.

| Material | E (GPa) | $\sigma_y$ (MPa) | $\varepsilon_t$ (%) |
|---|---|---|---|
| PP | 27.35 ± 0.64 | 27.16 ± 0.81 | 179.31 ± 5.14 |
| PP/WF-5901 | 47.02 ± 1.61 | 22.50 ± 0.93 | 3.140 ± 0.31 |
| PP/WF-5951 | 49.96 ± 2.38 | 21.43 ± 1.08 | 2.12 ± 0.22 |
| PP/WF-0218 | 45.31 ± 1.29 | 21.49 ± 0.53 | 3.86 ± 0.37 |

The use of coupling agents promotes a very noticeable effect on mechanical behavior. It is possible to highlight the increase in stiffness in PP/WF composites. However, the addition of coupling agents and WF strongly affected strength and ductility.

The elastic modulus of the composites increased up to approximately 82% relative to the Young's modulus of PP. The PP/WF-5951 and PP/WF-5901 composites presented stiffness values of 71.9 and 82.6% higher than neat PP, respectively. These values contrast with the increase in stiffness of the PP/WF-0218 composite, which was slightly lower (2–4%) than the other composites. This behavior could indicate a better intermolecular performance of the PP chains with the grafted copolymers compared to the PP/PE masterbatch coupling of CR0218.

Furthermore, some investigations indicate that the variations in the elastic modulus of the WPCs relative to the neat polymers imply a reinforcing effect on the part of the WF. The WF's rigidity restricts the polymer chains' molecular mobility [38]. Nevertheless, the polymer–reinforcement interaction, as well as the crystallization capacity of the polymer, should be factors to consider in the mechanical analysis of WPCs. On the other hand, tensile strength decreased up to 22% in PP/WF composites. The reduction in the yield strength values with the addition of WF could be related to poor adhesion of the wood flour with the polymer matrix, which could be associated with excessive filler or low coupling agent content.

The nominal strain of PP/WF composites was drastically decreased by up to 98% compared to neat PP, due to a possible agglomeration of the filler in the polymer matrix that prevents the alignment and unfolding of the crystals during the tensile test, causing the sudden failure observed in Figure 2b beyond the yielding point.

Figure 3 shows the tensile specimens corresponding to neat PP (white specimen) and PP/WF composites (dark specimens). The PP/WF specimens clearly show the absence of a localized deformation, or necking, which indicates that the WF filler hinders the mobility and displacement of the polymeric chains and interrupts the yielding of the PP.

The incorporation of coupling agents and wood flour in the PP affected the mechanical performance of the PP/WF composites. Some works reported that the use of graft coupling agents promotes a decrease in the molecular weight of PP [55]. On the other hand, the intrinsic nature of wood flour particles restricts deformability and increases stiffness, which implies a notable reduction in nominal strain in PP/WF composites [38].

As previously mentioned, coupling agents and WF fillers can induce affinity and reinforcing effects in PP, respectively. Therefore, it is necessary to elucidate the impact of WF and coupling agents on the crystallinity of PP/WF composites. Figure 4 shows the DSC endotherm curves for the first and second heating scans corresponding to the neat PP as received and the composite pellets prepared in extrusion. The DSC curves at the left correspond to the first heating; the DSC curves at the right represent the second heating scans.

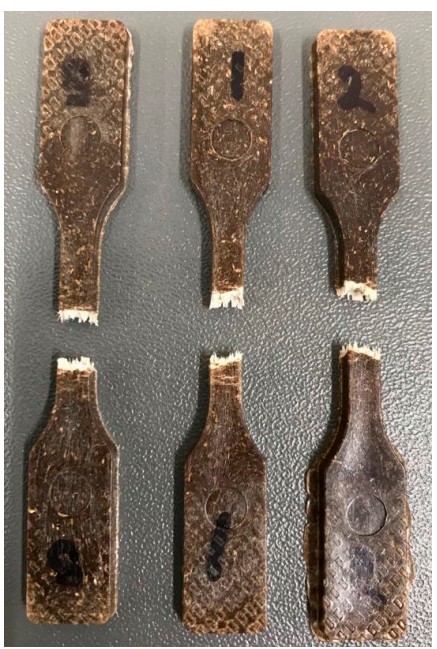

**Figure 3.** Tensile-broken specimens corresponding to PP/WF-5901, PP/WF-5951, and PP/WF-0218 composites.

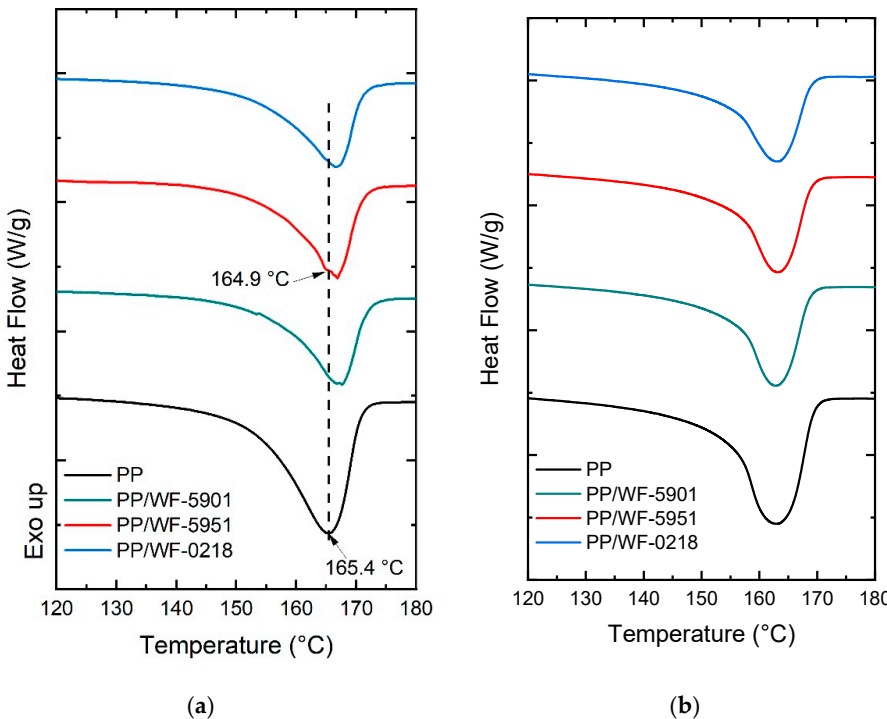

(**a**) (**b**)

**Figure 4.** DSC heating scan for (**a**) the first heating and (**b**) the second heating scans.

Appreciating broader endothermic signals for the first and second heating is possible. However, the most significant differences occurred in the first heating (Figure 4a). During the first heating, the endothermic signals of the PP/WF composites were slightly shifted toward higher temperatures compared to the endotherm of PP. In addition, only one melting peak can be observed for neat PP at 165.4 °C, while two diffuse melting peaks can be observed for PP/WF composites, which are more noticeable for PP/WF-5901 and PP/WF-5951. One of the melting peaks of PP/WF-5951 appears at a lower temperature (164.9 °C) than neat PP, which suggests the formation of less-perfect crystalline popula-

tions (grafting branches) [56,57] or lower molecular weights [38]. On the other hand, the displacement of the endothermic signals can be associated with a molecular orientation of PP chains during the extrusion process.

The DSC test consists primarily of a first heating scan, a controlled cooling, and a second heating scan. During the first heating scan, an erasure of the material's thermomechanical history occurs, showing traces of the transformation process, such as molecular orientation. Controlled cooling allows evaluation of the ability of the polymer to crystallize. The second heating melts the crystals formed during the cooling scan, and thermal phenomena such as effective crystallization, nucleation, degradation, and chain scission are revealed. Figure 4b presents the endotherms of the second heating scan. It can be seen that the endothermic signals are similar, and there are no significant variations in the position of the peaks of the neat PP and PP/WF composites. The results allow us to consider that the extrusion process led to molecular orientation without degrading the polymer. The similar shapes of the endotherms of the second heating scan could indicate the formation of homogeneous crystalline populations [56,57].

Some authors [58] have remarked that the display of an endothermic melting peak of neat PP is associated with melting the α-crystal. The PP/WF composites modified with coupling agents exhibited two diffuse endothermic peaks, which may correspond to the fusion of the β-form and the original α-form [58]. The diffused definition of these melting peaks would indicate the insufficient formation of crystals, which rearranged in the second heating scan, eliminating the β-form. For that reason, the second heating only reveals the formation of a melting peak for neat PP and PP/WF composites.

The thermal parameters obtained from DSC curves corresponding to the first and second heating scans are summarized in Table 2.

**Table 2.** Thermal parameters of PP and PP/WF composites.

| | First Heating | | Second Heating | |
|---|---|---|---|---|
| **Material** | **Tm (°C)** | **Xm (%)** | **Tm (°C)** | **Xm (%)** |
| PP | 165.4 | 17.8 | 162.8 | 21.0 |
| PP/WF-5901 | 167.7 | 23.4 | 162.6 | 28.7 |
| PP/WF-5951 | 166.9 | 23.5 | 163.4 | 28.7 |
| PP/WF-0218 | 166.7 | 25.2 | 163.2 | 28.3 |

As expected, higher melting temperatures for PP/WF composites than neat PP were detected in the first heating. However, in the second heating scan the melting temperatures were notably lower than the first heating temperatures for all materials evaluated. In addition, the melting temperatures do not show significant variations between the neat PP and the PP/WF composites, possibly due to the rearrangement of oriented chains during the melt-blending process. Similar melting points confirm no evidence of thermal degradation.

The crystallinity of neat PP turned out to be lower than the PP/WF composites for both the first and second heating. In the second heating scan, the degree of crystallinity of the neat PP and PP/WF composites increased, confirming a rearrangement of the oriented chains during crystallization.

Crystallinity could be favored in PP/WF composites due to the coupling agents, the nucleating effect promoted by WF particles, or a combined effect of both. However, the WF particles' excessive nucleation contribution would limit their solubility with molten PP [58]. Some authors [38,59] have related the increase in crystallinity produced by the coupling agents, associating a lower average molecular weight with grafting. Other authors [58] have pointed out that the presence of WF favors the formation of crystals around the WF, probably due to hydrogen bonding between WF and coupling agents. Both would favor an increase of the crystallinity of the PP/WF composites.

Returning to the mechanical properties, in order to better understand the mechanical behavior of the PP/WF composites observed in Figure 2, it is necessary to relate the effect of

the processing methods and crystallinity. Therefore, DSC tests were performed on samples taken directly from the gauge length of the ASTM type V specimens to evaluate crystalline changes influencing the composites' mechanical behavior. The endotherms corresponding to the first and second heating are observed in Figure 5. The thermal parameters are presented in Table 3.

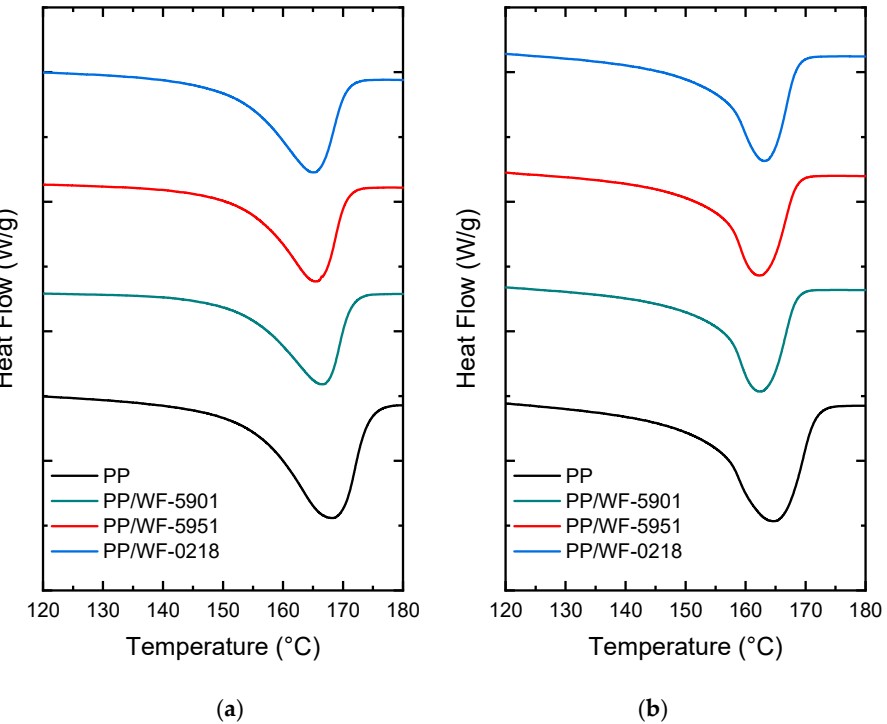

**Figure 5.** DSC heating scan for (**a**) the first heating and (**b**) the second heating scans from the gauge length of the injected tensile specimens.

**Table 3.** Thermal parameters of PP and PP/WF composites from the gauge length of tensile specimens.

| Material | First Heating | | Second Heating | |
|---|---|---|---|---|
| | Tm (°C) | Xm (%) | Tm (°C) | Xm (%) |
| PP | 168.3 | 16.8 | 164.7 | 20.8 |
| PP/WF-5901 | 166.5 | 29.0 | 162.5 | 29.3 |
| PP/WF-5951 | 165.5 | 26.5 | 162.3 | 28.3 |
| PP/WF-0218 | 165.1 | 23.2 | 163.2 | 29.1 |

It can be seen that the shape of the endothermic peaks of the first and second heating scans are very similar, so the temperature at which the tensile specimens are injected eliminates the thermomechanical history of the pellets produced during the melt-blending process.

Interestingly, the melting temperature of the neat PP from the first and second heating was the highest of all the DSC analyses performed in this work, 168.3 °C and 164.7 °C, respectively. The melting temperatures of the composites remained practically without relevant changes, both for extrusion and injection molding. These data would indicate the preferred orientation of the neat PP chains during the injection molding and the WF particles restricting the ordering of polymer chains during the transformation process.

On the other hand, the crystallinity values of neat PP are similar concerning the melt-blending process and injection molding and remain lower than those obtained from PP/WF composites. The crystallinity of the injected composite specimens became higher than that

of the extruded pellets. This would indicate that the composites maintain their ability to crystallize.

Polypropylene is a polymer that contains very low intermolecular interaction forces between chains, so its mechanical strength derives mainly from its crystallinity. In this sense, as the crystallinity decreases, the resistance can decrease unless the adhesion between phases compensates for the loss of crystallinity [60]. This may be because WF particles provide a surface for PP nucleation.

One way to understand whether the coupling agents achieve compatibility between the polymer chains and the WF particles is through SEM observations. SEM micrographs of the failure surfaces corresponding to the PP/WF composites are shown in Figure 6.

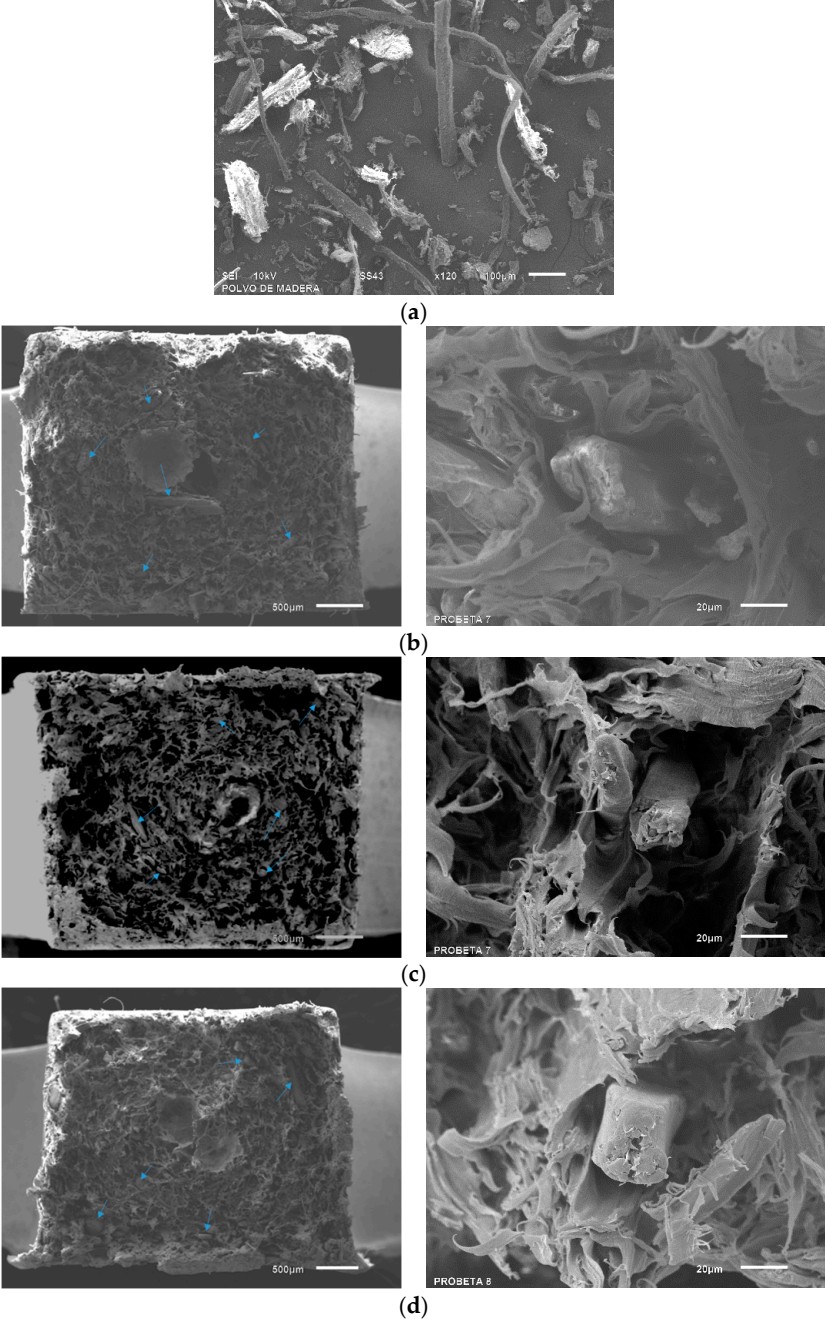

**Figure 6.** SEM micrographs of (**a**) WF and failure surfaces of tensile specimens corresponding to (**b**) PP/WF-5901, (**c**) PP/WF-5951, and (**d**) PP/WF-0218. Lower magnifications are to the left and higher magnifications to the right.

Figure 6a presents the WF particles. It can be seen that the WF particles are heterogeneous in their dimensions and shapes.

Figure 6b,c correspond to the observations made to the PP/WF composites at low and high magnifications. The PP/WF composites showed similar surfaces regardless of the organic modifier used.

Low magnifications allow us to appreciate a rough and tearing surface, even being able to visualize WF particles on the surface of the composites. Higher magnifications allow for distinguishing more specific features of the WF particles in the PP matrix.

The PP/WF composites show discontinuous and rough surfaces due to the presence of the WF particles, as indicated with blue arrows in Figure 6b–d. The discontinuous surfaces would indicate an insufficient adhesion between the phases, which promotes the reduction of the stress transfer observed during the tensile test and decreases the composites' resistance and ductility [55]. Therefore, the mechanical behavior of PP/WF composites would be strongly influenced by the crystallinity of the composites, being favored by the PP–coupling agent interaction. For their part, the WF particles would act as a filler and not as a reinforcement, increasing the rigidity and reducing the molecular displacements of the polymer so that the Young's modulus increases and the deformation decreases significantly.

## 4. Conclusions

PP/WF composites incorporating graft coupling agents and masterbatch were developed to evaluate their effect on mechanical parameters in this work. The composites were prepared using the melt-blending process. The PP showed a typical mechanical behavior, with a marked neck and a wide deformation capacity. The mechanical behavior of the PP/WF composites revealed an elastic region followed by a termination after their yield point, shortening the stress–strain curves and reducing their ductility. Graft coupling agents had a better intermolecular performance with PP than masterbatch coupling agents. Coupling agents improved the crystallinity of PP in favor of tensile strength. The SEM observations showed insufficient adhesion between the WF and the PP, which promotes the reduction of stress transfer. WF particles acted as fillers that increased the stiffness and reduced the ductility of the composite materials. The results obtained allow us to continue with the evaluation of these materials and to assess the influence of humidity on them. The ability of coupling agents to restrict water uptake will allow the results obtained to be contrasted in this future work.

**Author Contributions:** Conceptualization, resources, writing—original draft preparation, and investigation, E.A.F.-U.; formal analysis and investigation, R.R.-A.; writing—review and editing, E.A.F.-U.: methodology, visualization, and investigation, C.Z.-P.; review, C.S.-A. All authors have read and agreed to the published version of the manuscript.

**Funding:** This work was supported by the Secretary of Public Education (SEP) and the National Council of Science and Technology (CONACYT) through the Basic Science Fund (SEP-CONACYT Ciencia Basica), grant number 257458, and in collaboration with ADAPT.

**Data Availability Statement:** This study does not report any data.

**Acknowledgments:** The authors want to thank the valuable support of Juan Manuel Gonzalez Carmona and Antonio Banderas Hernandez for assistance with SEM micrographs also to Ricardo Alberto Lozada Loyola with Mechanical test.

**Conflicts of Interest:** The authors declare no conflict of interest.

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
