# Peer review of "The Role of Coupling Agents in the Mechanical and Thermal Properties of Polypropylene/Wood Flour Composites"

_2673-6209, doi:10.3390/macromol3010006_

Round 1

Reviewer 1 Report

Dear Authors,

the article is interesting and deals with current issues. I believe that gain knowledge is important from pratctical point of view. However, I have some doubts. In my opinion that paper need more scientific sound.

Please consider my comments below:

L46 Do you believe that "Wood-plastic composites (WPC) are environmentally friendly hybrid composites"? Even some people think that plastic composites can be environmentally friendly, they are still synthetic composites and further utilisation after its life cycle.

L78 "The consulted literature" it does not sound proper

L105 When you introduced the aim of the work, there should be also pointed when desired changes are needed - in what kind of finished products. And one more important thing, in what sense your solution will be better than others? The readers shoul be informed of that.

Materials description - there is no grammar here. Consider using table 

Table 1 Some changes are presented, but it is necessary to add statistical evoluation to confirmed that they are differences and to make proper conlusion.

Table 2 title need to be rather "Presentation ..."

L317 "The crystallinity of the neat PP was calculated by the following equation ..." it is part for methodology description

There are two Figure 5

Conclusion are two general. You did a big work. But, you need organize the article, analyze the results in terms of their statistical significance and then modify the discussion and formulate final conclusions. Please indicate the practicality of the results there.

Author Response

Reviewer 1

Dear Authors,

the article is interesting and deals with current issues. I believe that gain knowledge is important from pratctical point of view. However, I have some doubts. In my opinion that paper need more scientific sound.

The authors appreciate the comments that undoubtedly enrich the manuscript. Thank you

Please consider my comments below:

L46 Do you believe that "Wood-plastic composites (WPC) are environmentally friendly hybrid composites"? Even some people think that plastic composites can be environmentally friendly, they are still synthetic composites and further utilisation after its life cycle.

The authors agree with the Reviewer. The paragraph has been modified. Thanks

L78 "The consulted literature" it does not sound proper

The authors appreciate the recommendation, this sentence has been removed.

L105 When you introduced the aim of the work, there should be also pointed when desired changes are needed - in what kind of finished products. And one more important thing, in what sense your solution will be better than others? The readers shoul be informed of that.

The evaluation of the mechanical properties of PP/WF composites using organic modifiers added during their mixing in a melt-blending process provides useful information to know the performance of these materials and elucidate the effects of thermal degradation or molecular compatibility. This is specified in more detail throughout the article. The authors modified the text. Thanks

Materials description - there is no grammar here. Consider using table 

The authors consider this option. However, it is not possible to use a table to describe the materials as they are all of a different nature. However, the authors modified the paragraph. Thank you

Table 1 Some changes are presented, but it is necessary to add statistical evoluation to confirmed that they are differences and to make proper conlusion.

The main differences can be seen clearly in the values of the table, comparing the composites without and with modifiers. Subsequently, the statistical values are described in the text of the manuscript.

Table 2 title need to be rather "Presentation ..."

The title was modified. Thank you

L317 "The crystallinity of the neat PP was calculated by the following equation ..." it is part for methodology description

This part was added to the methods section. Thank you

There are two Figure 5

Thank you for this observation. The Figures have now the correct numbering

Conclusion are two general. You did a big work. But, you need organize the article, analyze the results in terms of their statistical significance and then modify the discussion and formulate final conclusions. Please indicate the practicality of the results there.

The authors appreciate the Reviewer's comment. However, the authors consider that the conclusions of this work are very specific but allow continuing with the studies to compare other effects and enrich the research on these composites. Thank you very much for your recommendations, which undoubtedly helped to improve the manuscript.

Reviewer 2 Report

the article analysis the effect of few coupling agents on the performance of wood-based composites. I have prepared few remarks which I hope will help to improve the quality of the article:

1. It would be easier to compare tensile strength results in Figure 2 if the y axis had the same scaling in both graphs.

2. Please use the same accuracy when presenting numerical values in Table 1, e.g. 27.16 and 22.5 should be written as 27.16 and 22.50 or 27.2 and 22.5 MPa.

3. Calculation formuales should be presented in methods part but not in Results and Discussion part.

4. SEM images in Figure 5 does not contain the magnification.

5. There are some publications proving that the use of coupling agents reduce water uptake of particle-based composites. Reviewers suggest adding some additional results regarding water absorption. Also, in order to show the impact of coupling agents on the mechanical performance of samples, it is necessary to do SEM images of fractured surface. Additionally, I would suggest testing compressive strength properties also

Author Response

Reviewer 2

the article analysis the effect of few coupling agents on the performance of wood-based composites. I have prepared few remarks which I hope will help to improve the quality of the article:

The authors appreciate the comments that undoubtedly enrich the manuscript. Thank you

  1. It would be easier to compare tensile strength results in Figure 2 if the y axis had the same scaling in both graphs.

Thank you for this observation. The y-axis was modified.

  1. Please use the same accuracy when presenting numerical values in Table 1, e.g. 27.16 and 22.5 should be written as 27.16 and 22.50 or 27.2 and 22.5 MPa.

Thank you for this observation, the values were written correctly.

  1. Calculation formuales should be presented in methods part but not in Results and Discussion part.

The Authors agree with the Reviewer’s comment. The formulas are now in the methods section. Thank you.

  1. SEM images in Figure 5 does not contain the magnification.

The SEM images were modified.

  1. There are some publications proving that the use of coupling agents reduce water uptake of particle-based composites. Reviewers suggest adding some additional results regarding water absorption. Also, in order to show the impact of coupling agents on the mechanical performance of samples, it is necessary to do SEM images of fractured surface. Additionally, I would suggest testing compressive strength properties also

The authors are grateful for the Reviewer's comments. The results obtained allow us to continue with the investigation and the authors are analyzing these results, which will complement the investigation.

The authors appreciate the comments and recommendations of the Reviewer, Thank you

Reviewer 3 Report

Dear Authors,
I found your manuscript valuable and up-to-date due to readers' high potential interest in the WPC field. However, please, consider applying the following remarks:
- line 16-17, when providing the hydrophilic phase and polymer matrix, please indicate which is wood powder and which is polymer matrix (since wood is also considered as a polymer)
- line 48, when saying that WPC does not harm the environment, please, go deeper and explain this, since WPC contains petrochemical-based thermoplastic polymers
- line 182-183 - please unify the font size
- line 390 (Figure 5) - please indicate the crucial areas/points directly on the pictures, as mentioned in the description provided in lines 395-397

Best regards!

Author Response

Reviewer 3

Dear Authors,
I found your manuscript valuable and up-to-date due to readers' high potential interest in the WPC field.

The authors appreciate the comments that undoubtedly enrich the manuscript. Thank you

However, please, consider applying the following remarks:
- line 16-17, when providing the hydrophilic phase and polymer matrix, please indicate which is wood powder and which is polymer matrix (since wood is also considered as a polymer)

Thank you for the observation. The paragraph indicates this recommendation.

- line 48, when saying that WPC does not harm the environment, please, go deeper and explain this, since WPC contains petrochemical-based thermoplastic polymers

Yes, the Authors agree with this observation and the paragraph was modified. Thank you.

- line 182-183 - please unify the font size

The font size was unified

- line 390 (Figure 5) - please indicate the crucial areas/points directly on the pictures, as mentioned in the description provided in lines 395-397

The authors appreciate the comments. SEM images were modified. Thank you

Best regards!

Round 2

Reviewer 2 Report

Authors have answered all my questions. I do suggest publication of this article.